# Aromatic Characterization of Mangoes (*Mangifera indica* L.) Using Solid Phase Extraction Coupled with Gas Chromatography–Mass Spectrometry and Olfactometry and Sensory Analyses

**DOI:** 10.3390/foods9010075

**Published:** 2020-01-09

**Authors:** Haocheng Liu, Kejing An, Siqi Su, Yuanshan Yu, Jijun Wu, Gengsheng Xiao, Yujuan Xu

**Affiliations:** Sericulture & Agri-Food Research Institute Guangdong Academy of Agricultural Sciences/Key Laboratory of Functional Foods, Ministry of Agriculture and Rural Affairs/Guangdong Key Laboratory of Agricultural Products Processing, Guangzhou 510610, China; AnsisHC@163.com (H.L.); ankejing@gdaas.cn (K.A.); lijun@gdaas.cn (S.S.); yuyuanshan@gdaas.cn (Y.Y.); wujijun@gdaas.cn (J.W.); gshxiao@yahoo.com.cn (G.X.)

**Keywords:** mango, volatile compounds, frequency detection (FD), order-specific magnitude estimation (OSME), odor activity value, sensory analysis

## Abstract

Mangoes (*Mangifera indica* L.) are wildly cultivated in China with different commercial varieties; however, characterization of their aromatic profiles is limited. To better understand the aromatic compounds in different mango fruits, the characteristic aromatic components of five Chinese mango varieties were investigated using headspace solid-phase microextraction (HS-SPME) coupled with gas chromatography-mass spectrometry-gas chromatography-olfactometry (GC-MS-O) techniques. Five major types of substances, including alcohols, terpenes, esters, aldehydes, and ketones were detected. GC-O (frequency detection (FD)/order-specific magnitude estimation (OSME)) analysis identified 23, 20, 20, 24, and 24 kinds of aromatic components in Jinmang, Qingmang, Guifei, Hongyu, and Tainong, respectively. Moreover, 11, 9, 9, 8, and 17 substances with odor activity values (OAVs) ≥1 were observed in Jinmang, Qingmang, Guifei, Hongyu, and Tainong, respectively. Further sensory analysis revealed that the OAV and GC-O (FD/OSME) methods were coincided with the main sensory aromatic profiles (fruit, sweet, flower, and rosin aromas) of the five mango pulps. Approximately 29 (FD ≥ 6, OSME ≥ 2, OAV ≥ 1) aroma-active compounds were identified in the pulps of five mango varieties, namely, γ-terpinene, 1-hexanol, hexanal, terpinolene trans-2-heptenal, and *p*-cymene, which were responsible for their special flavor. Aldehydes and terpenes play a vital role in the special flavor of mango, and those in Tainong were significantly higher than in the other four varieties.

## 1. Introduction

Mango (*Mangifera indica* L.), a native crop of South Asia, is a member of the *Anacardiaceae* family [1] and has been historically grown for more than 4000 years [2], thereby earning the title “king of fruits”. It is recognized as one of the most popular fruits around the world, with the highest rates of production, marketing, and consumption [3,4]. Among tropical fruits, mango is the second most common crop involved in international trade, following banana. Global mango production has been estimated at 50.65 million tons, with China being the second largest mango-growing country, 2017 production reaching close to 4.94 million tons [5]. Common mango cultivars in China featuring specific regional characteristics include Guifei, Hongyu Jinmang, Qingmang, and Tainong.

Aroma is a major factor that influences the quality and consumer acceptance of mango products. Investigating various aromatic components would improve our understanding and facilitate controlling critical quality parameters that could influence mango processing. Hundreds of compounds have been characterized in various mango cultivars, which mainly include aldehydes, alcohols, esters, ketones, and terpenes [6]. However, previous studies have mainly focused on the volatiles in various mangoes in China [7,8], and a few volatiles were detected because of their odor threshold. Thus, scientific information relating to aromatic constituents as well as sensory characteristics of various mango cultivars is limited. Therefore, an in-depth investigation is required to identify the volatile or aromatic components of various mango cultivars in China.

To determine the aromatic components of mango, simultaneous solvent-assisted flavor evaporation (SAFE), solid phase microextraction (SPME), and distillation and extraction (SDE) have been employed in food aroma extraction [9,10,11,12]. The procedures of SDE and SAFE isolate aromatic compounds from food matrices using organic solvents [13]; however, these methods are highly laborious, time-consuming, and entail preconcentration of extracts. Unlike well-established protocols, SPME has been extensively used in the preparation of volatile and semi-volatile compounds from various types of samples [14]. This technique was developed more than two decades ago and is a rapid, simple, sensitive, and solvent-free technique for the analysis of volatile organic compounds (VOCs) [15]. Coupling of the methods of aroma extraction with the GC-MS/O technique, in particular, headspace solid-phase microextraction (HS-SPME) for extraction, together with detection frequencies (FD) and order-specific magnitude estimation (OSME) for GC-MS/O, creates a highly reliable method of identifying potent odorants.

Thus, to fully understand the aromatic compounds present in typical mango varieties, this study conducted the following studies: (1) identification and quantification of volatiles of different types of mangoes by HS-SPME-GC-MS; (2) discrimination of the major aroma-active compounds in various types of mangoes using combined GC-O detection (FD, OSME) and odor activity value (OAV); and (3) validation of the sensory differences using quantitative descriptive analysis (QDA). In summary, objectives of this investigation were to reveal the major aromatic compounds of mangoes in China.

## 2. Materials and Methods 

### 2.1. Samples Preparation 

This study used seven mature samples of the cultivars Jinmang (JM), Qingmang (QM), Hongyu (HY), Guifei (GF), and Tainong (TN), which were purchased from the regional market in Guangzhou (133.35° N, 23.12° E), Guangdong Province (April 2019). The samples were shipped to the laboratory and kept at 25 °C until complete maturity (2 days). Full maturity and maturity of the mangoes were based on fruit color (green to yellow-orange or red, except for the green lawn, Appendix A), odor (sweet scent), and hardness (pulp hardness index changed from 5.28 to 4.32 N). Finally, the mango samples with the same maturity were washed, and the peeled pulp was immediately frozen in liquid nitrogen, and then stored at −80 °C for further studies.

### 2.2. Chemicals

Humulene, 2-penten-1-ol, 2-hexen-1-ol, (E)-3-hexen-1-ol, 1-hexanol, p-cymen-8-ol, 2-vinyloxy)-ethanol, 3-methyl-1-butanol, 1,3,8-*p*-menthatriene, allo-ocimene, 2-carene, α-phellandrene, 3-carene, terpinolene, 1,3-cyclohexadiene, 1-methyl-4-(1-methylethyl), isovaleraldehyde, 3-hexenal, 2,4-dimethyl-benzaldehyde, heptanal, *trans*-2-heptenal, trans-2-pentenal, 1-nonanal, decanal, citral, 1-penten-3-one, 2-cyclohepten-1-one, 3-methylcyclohex-3-en-1-one, 6-methyl-5-hepten-2-one, ethyl-propionate, ethyl cyclopropanecarboxylate, ethyl crotonate, isoamyl acetate, tetraethyl orthosilicate, ethyl butyrate, and γ-octanoic were purchased from TCI (Tokyo, Japan); and linalool, α-pinene, β-pinene, β-myrcene, D-limonene, *p*-cymene, β-ocimene, hexanal, γ-terpinene, and *trans*-2-hexenal were purchased from Sigma (St. Louis, MO, USA). All of the chemical standards were of GC quality. 

### 2.3. HS-SPME-GC-MS

Based on previous studies, the optimized SPME experimental conditions were established [16,17,18]. Approximately 5.0 g of the juice with 1.5 g of NaCl were blended in a 15 mL vial tightly capped with a PTFE-silicon septum at a stirring speed of 80 rpm. The flavor compounds in mango pulp are formed during equilibrium and during the extraction process. Therefore, the extraction temperature was set at 40 °C. After the vial containing the sample was equilibrated at 40 °C for 10 min on a heating platform agitation, the pretreated (conditioned at 270 °C for 30 min) SPME fiber (50/30 μm DVB/Carboxen/PDMS, Supelco, Bellefonte, PA, USA) was then inserted into the headspace, and extraction was performed for 30 min with continued heating and agitation. Afterward, the fiber was withdrawn and instantly introduced to the GC for desorption and analysis. 

GC-MS analysis was performed on an Agilent 7890 (Agilent Technologies, Palo Alto, CA, USA) gas chromatography and Agilent 5977 mass selective detector. Samples were separated using both HP-5 and DB-WAX (both 30 m × 0.25 mm i.d., 0.25 mm film thickness, Agilent Technologies, Palo Alto, CA, USA). Helium was used as carrier gas at a flow rate of 1.7 mL/min, and the GC inlet was set in the split-less mode. The injector temperature was 250 °C. The temperature program was from 40 °C (2 min hold) to 160 °C at 4 °C/min and finally raised to 280 °C at 50 °C/min. Then, electron ionization mode (EI) was used with a 70 eV ionization energy. The ion source temperature was 230 °C, and the mass range was from *m*/*z* 35 to 450. The volatile compounds were determined by authentic standards, retention indices (RI), and NIST 14.0 library. The retention indices (RIs) of compounds were determined via sample injection with a homologous series of straight-chain alkanes (C6–C30) (Sigma Aldrich, St. Louis, MO, USA).

### 2.4. Identification of Aroma-Active Compounds by GC-O

The odorant compounds were analyzed using a sniffing port (ODP3, Gerstel, Germany) coupled with a GC-MS (7890B–5977B, Agilent Technologies, Inc.). Upon exiting the capillary column, the effluents were divided to a ratio of 3:1 (by volume) into a sniffing port as well as an MS detector using an Agilent capillary flow technique. The transfer line directed to the GC-O sniffing port was set at a temperature of 270 °C. The GC-MS settings were similar to those described earlier. Aroma extraction was conducted by four highly skilled personnel (in an alternate order of 50 min intervals) using reference compounds. All personnel were extensively trained on the GC-O technique for at least 90 h.

Frequency analysis was conducted by four trained sensory panelists (i.e., two males and two females). Retention time and odor quality, together with substance detection, were recorded. Frequency analysis was performed in duplicate by every panelist. Odorants with an FD ≥2 (determined by at least two analysts) were considered to have potential aroma activity [19].

The OSME reflected the aromatic intensity of the stimulus that was based on a five-point scale that ranged from 0 to 5, where 0 = none, 3 = moderate, and 5 = extreme. Every sample was sniffed thrice by each panelist, and then the average aromatic intensity values were calculated. If the panelists did not utilize a similar attribute for an aroma that was eluted by GC, the analysis was repeated, and only the descriptors used for the same aroma were included in the analysis [20].

### 2.5. Quantitative Analysis of Aromatic Compounds

Quantitative data on the identified compounds were gathered by calculating their relative quantitative correction factors (RQCFs) with the “single-point correction method”, which is similar to the standard addition method. Similar GC conditions as described above were used in GC-MS analysis, with solvent delay time set at 3 min to prevent the solvent in the standard solutions to reach the filament. The method of obtaining RQCFs consisted of the following: 5.0 g of mango pulp was analyzed using SPME-GC-MS, resulting in an ion peak area for each identified compound. A similar volume of mango pulp with defined amounts of various authentic and internal standards (to avoid run-to-run variations) was then analyzed to generate a new quantifying ion peak for each detected compound and quantitative correction factor for the internal standard. The RQCF of each volatile was generated using the following equation:(1)$$f_{\bar{i}}^{\prime }=\frac{f_{wi}}{f_{ws}}=\frac{m_{i}/A_{i}}{m_{s}/A_{S}}=\frac{A_{S}m_{i}}{A_{i}m_{s}},$$
where *f_i_′* was the RQCF of a detected compound (*i*); *m_s_* and m_i_ were the respective known contents of authentic (*i*) and internal standard (*s*); *A_s_* was the peak area of the quantifying ions of (*s*); and *A_i_* was the peak area of the quantifying ions of (*i*) before and after the addition of the standard solution to the juice sample.

To determine the amounts of the identified volatiles in mango pulp, approximately 5.0 g of juice per volume containing a similar amount of internal standard as that of the calculated RQCF was prepared and used in GC-MS analysis. The concentration was computed using the equation:(2)$$m_{i}=f_{i}^{\prime }\times A_{i}\times \frac{m_{s}}{A_{s}}$$
where m_i_ was the amount of compound (i); *m_s_* was the known amount of (s), *A_i_* and *A_s_* were the respective peak areas of the quantifying ions of detected compound (i) and internal (s) standards; and *f*_i_′ was the RQCF of (i). The peak area of the quantifying ion of every component in selected ion chromatograms was assessed in triplicate, and the average value was computed. Then, the concentration of every identified volatile in mango pulp is described in nanograms per milliliter of juice [19].

### 2.6. Odor Activity Value (OAV)

OAV pertains to the concentration of the odor divided by its threshold in water. Compounds with an OAV ≥1 were considered as major contributors to the aromatic profile of each sample [21].

### 2.7. Sensory Panel and Aroma Profile Analysis 

Aromatic profiling was performed using descriptive sensory analysis, as earlier described [22]. The mango pulps were analyzed by a highly skilled panel of 10 members consisting of five males and five females. Prior to quantitative descriptive analysis, 50 mL of mango pulp was placed in a 100 mL cubage of a plastic cup with a Teflon lid, which was handed over to a panelist in the laboratory without peculiar smell at a temperature of 25 °C. Then, the panelists assessed the aromatic profile of the mango pulp using three preliminary sessions (each spent approximately 2 h), until all of them attained a consensus as to the degree of aromatic flavor. Then, the organoleptic characteristic descriptors were assessed using eight sensory features (i.e., “overall aroma”, “tropical fruit”, “citrus”, “floral”, “fresh”, “rosin”, “honey/sweet”, and “green”) to evaluate aroma negative and positive mango pulp features. The descriptors were described as the following odors: linalool for the “floral” descriptor, β-phellandrene for “citrus”, butyl acetate for the “fruity” descriptor, β-damascenone for the “honey/sweet” descriptor, phenylacetaldehyde for “fresh”, (E)-2-hexenal for “green and grassy”, and terpinolene for “rosin”. The complete profile of each sample was randomly assessed in triplicate for each treatment. The assessors then rated the odor intensities a seven-point scale ranging from 0 to 3, with 0 as not perceivable, 1 as weak, 2 as significant, and 3 as strong. The results were then averaged for every odor note and plotted using a spider web diagram.

## 3. Results

### 3.1. Sensory Analysis of Five Mango Cultivars

Figure 1 shows that the five mango pulps had similar aromatic intensities, and these can be divided into six aroma attributes, including tropical fruit, flower aroma, sweet aroma, green grass aroma, green melon aroma, wood aroma, and rosin aroma, although to different extents. SPSS was used to distinguish differences between the mango samples through sensory evaluation scores (Appendix A). In most cases, the five mango pulps exhibited significant differences in intensity of fruit aroma (*p* < 0.05), with some exceptions for “tropical fruit” and sweetness between JM and TN, tropical fruit between GF and QM, green between JM and HY, melon between GF and TN, and wood between JM and GF, where no significant difference was observed (*p* > 0.05). In general, innate differences among the mango varieties significantly influenced the intensities of most of the key sensory attributes. 

The overall (decreasing) order of flavor scores was: TN > GF > HY > JM > QM. The intensity of fruit aroma was highest in both GF and QM, followed by TN and JM. The intensity of flower aroma was highest in both GF and HY, followed by QM and JM. TN had the lowest. GF and HY had the highest intensity of sweet aroma, followed by TN, QM, and JM. TN had the highest intensity of green grass aroma, followed by GF and QM. The intensity of green grass aroma in JM and HY was almost identical. The intensities of green melon aroma in QM, GF, and TN were markedly higher than those of JM and HY. TN had the highest intensity of green melon aroma. In addition, the intensity of wood aroma in GF and TN was higher than in JM, QM, and HY. TN had the highest intensity of wood aroma. 

In general, GF had the strongest flower and sweet aromas, HY had the strongest fruit aroma, and TN had the strongest green grass, green melon, wood, and Rosin aromas. According to Bonneau [9], different mango varieties significantly influenced the intensities of key sensory attributes of mango, which was due to the different amounts of aroma-active components in mango.

### 3.2. Comparasion of the GC-MS Results of Five Mango Cultivars

The MS and RI results preliminarily identified 47 volatile compounds (Figure 2). There were 25, 24, 24, 29, and 23 volatile compounds in JM, QM, GF, HY, and TN, respectively. These were generally composed of alcohols, alkenes, aldehydes, esters, ketones, and ethers. Most of the volatiles found in this study were similar to those in the findings of previous studies [1,6,23]. Six compounds, namely, *p*-cymen-8-ol, 2-(vinyloxy)ethanol, 1-methyl-4-(1-methylethyl)-1,3-cyclohexadiene, 3-methyl butanal, 2-cyclohepten-1-one, and tetraethyl orthosilicate, were first observed in the volatile composition of mango, although those were not the major contributors to the mango aroma. This difference might be caused by habitat, maturity conditions, and aromatic extraction method. 

Figure 3 shows that the number of alkenes in the five mangoes was much higher than that of the other kinds of volatile compounds, indicating that alkenes were the most important volatile substances. In addition, the highest number of alcohols, terpenes, and aldehydes were found in JM, TN, and HY, respectively. The number of ketones in QM and HY was higher than in other mango varieties, whereas the number of esters in GF was higher than in other mangoes. Moreover, only nine volatile components, including one alcohol, one aldehyde, and seven terpenes, were identified in the five kinds of mango pulp. This showed that the volatiles of the five mango cultivars varied.

Full quantifications using standards were conducted for the volatile substances in the five mango varieties (Table 1 and Figure 4). In JM, the total amount of volatiles was 539.29 μg/kg; the content of terpinolene was the highest (147.07 μg/kg); then came 3-carene (103.63 μg/kg), 1-hexanol (55.83 μg/kg), e-3-hexen-1-ol (41.50 μg/kg), isoamyl alcohol (38.87 μg/kg), and *p*-cymene (27.43 μg/kg). Isoamyl alcohol (38.87 μg/kg) and 2-(vinyloxy) ethanol (3.49 μg/kg) were the unique volatile components. In QM, the total amount of volatiles was 662.92 μg/kg; the content of (e)-2-hexenal was the highest (180.60 μg/kg); then came e-3-hexen-1-alcohol (132.56 μg/kg), 3-carene (60.36 μg/kg), and terpinolene (59.42 μg/kg). The unique volatile components in QM were ethyl crotonate (9.83 μg/kg), e-2-nonenal (7.60 μg/kg), and 3-methyl-3-cyclohexene-1-one (0.62 μg/kg). In GF, the total amount of volatiles was 381.12 μg/kg; (e)-2-hexenal (84.03 μg/kg), trans-3-hexen-1-ol (76.82 μg/kg), and terpinolene (43.53 μg/kg) were the main volatile substances; the unique volatile component was isovaleraldehyde (1.77 μg/kg). In HY, the total amount of volatiles was 400.50 μg/kg; (e)-2-hexenal (85.64 μg/kg), hexanal (75.54 μg/kg), (e)-2-heptenal (65.65 μg/kg), 3-carene (27.67 μg/kg), and terpinolene (24.90 μg/kg) were the main volatile substances; HY is the most special kind of mango. Its unique volatile components were β-caryophyllene (2.18 μg/kg), (e)-2-heptenal (65.65 μg/kg), *p*-cymen-8-ol (12.23 μg/kg), ethyl propionate (6.39 μg/kg), 3-hexen-1-ol (6.59 μg/kg), humulene (2.93 μg/kg), β-caryophyllene (2.18 μg/kg), 2-cyclohepten-1-one (0.66 μg/kg), (e,z)-2,6-nonadienal (0.66 μg/kg), and ethyl cyclopropanecarboxylate (0.12 μg/kg). In the TN pulp, the total amount of volatiles was 1279.68 μg/kg; the content of terpinolene was the highest (811.61 μg/kg), followed by *p*-cymene (132.96 μg/kg), 3-hexenal (44.13 μg/kg), 3-carene (70.92 μg/kg), and phellandrene (30.17 μg/kg), with 3-hexenal being the unique volatile component. Based on the above analysis, the contents of volatile substances in the five cultivars of mangoes were different, and the total content of volatile substances in TN was the highest. And all five kinds of mangoes have unique volatile substances with individual aromatic components; especially for ruby, further analyses by GC-O are needed.

### 3.3. Identification of Key Aromatic Compounds in the Pulps of Five Mango Cultivars

Not all volatile substances in mango contribute to its aroma, and thus aromatic intensity identification was conducted to determine whether the high content or unique volatile components in mango contribute to its overall aroma. The frequency-of-detection (FD) method requires the evaluator to smell the same aromatic substance and record the peak time and aromatic properties of the aromatic substance. The more times the aromatic substance was detected, the greater the contribution to the overall aroma. The intensity (OSME) method was a GC-O detection method that was used to evaluate flavor contribution based on the odor intensity of the aroma substance. After the aromatic substances are separated by GC-MS, the method directly describes the change in odor intensity (measured) and the frequency of aroma attributes. It is considered to be the simplest and most effective GC-O analysis method because it is less time-consuming and less demanding on assessors. Thirty-three characteristic aromatic components were identified in the five mango pulps by GC-O combined with FD and OSME methods, including three alcohols, 10 aldehydes, 11 terpenes, three esters, and four ketones. The differences in aromatic components in the pulp of five mango cultivars were not significant. 

Further analysis, the FD analysis identified 23 aromatic substances (FD ≥ 2) in JM. According to the FD statistical data of each substance in the Table 2, one of the substances (FD = 8) was identified in all of the tests: γ-terpinene with citrus aroma. The substances with FD = 7 were phellandrene (citrus-like aroma) and γ-octanoic (floral, violet aroma). The substances with FD = 6 were ethyl cyclopropanecar-boxylate (fruit aroma) and terpinolene (rosin aroma). In QM, 18 aromatic substances with FD ≥2 were identified. According to the FD statistical data of each substance in the table, three kinds of substances (FD = 8) were identified in all of the tests, including 1-penten-3-one, 3-hexenal, and terpinolene. 1-penten-3-one had mushroom-like aroma, 3-hexenal had a grassy aroma, and terpinolene had a rosin-like aroma. The substance with FD = 7 was β-pinene, which has a green-grass-type aroma. The substances with FD = 6 were *p*-cymene (citrus and green aroma) and γ-terpinene (citrus and lemon-like aroma). In GF, 18 aromatic substances with FD ≥2 were identified. According to the FD statistical data of each substance in Table, 2 substances with FD = 8 were identified in all of the tests: ethyl cyclopropanecarboxylate (fruit aroma) and γ-terpinene (citrus and lemon aroma). The substances with FD = 7 were 3-hexenal (green grass aroma) and terpinolene (rosin-like aroma). The substance with FD = 6 was *p*-cymene (citrus and green aroma). In HY, 23 aromatic substances with FD ≥2 were identified. The substances with FD = 8 were ethyl cyclopropanecarboxylate (fruit aroma). The substances with FD = 7 were cis-2-penten-1-ol (grass and tea aroma), 3-hexenal (green grass aroma), and terpinolene (rosin-like). The substance with FD = 6 was *p*-cymene (citrus and green aroma). In TN, there were 24 aromatic substances with FD ≥ 2. The substances with FD = 8 were terpinolene and γ-terpinene, which mainly had a rosin citrus-like aroma. The substances with FD = 6 were ethyl cyclopropanecarboxylate, β-myrcene, 2-carene, phellandrene, and *p*-cymene. They had a fruity rosin and sweet aroma.

The OSME analysis showed that the aromatic intensities of the five mango pulps had significant differences (Figure 5). There were 23 kinds of aromatic components in JM, and the substances with the aromatic intensity greater than 2 were phellandrene (4.1), ocimene (2.33), γ-terpinene (2.3), and terpinolene (2.3). This suggests an overall aromatic profile of citrus, green grass, and flowers. There were 19 kinds of aromatic components in QM, and the substances with an aromatic intensity greater than 2 were 1-hexanol (2.12), methyl-3-methylcyclohex-3-en-1-one (2.67), β-phellandrene (2.1), and decanal (3) showing an overall aromatic profile of lemon, sweet, and green grass. Seventeen aromatic components were identified in GF, and the substances with aroma intensities greater than 2 were 1-hexanol (2.13), phellandrene (2.5), terpinolene (2.33), and p-menthol (2.25) showing an overall aroma profile of citrus and lemon, fresh green leaves, and green grass. There were 22 aromatic components in HY, and the substances with aromatic intensity greater than 2 were phellandrene (3), ocimene (2.5), α-pinene (2.2), γ-terpinene (2.5), terpinolene (2.2), and 1,3,8-*p*-menthatriene (2.2) showing an overall aroma profile of flower, citrus, green grass, and pine wood. TN had 22 aromatic components, including hexanal (2.2), 2-carene (2.2), phellandrene (3.53), *p*-cymene (8.7), γ-terpinene (2.5), terpinolene (2.8), and decanal (3) with aromatic intensity values greater than 2. This has an overall aroma profile of sweet, fruit, green grass, and pine wood.

There were no significant differences in the numbers of overall aromatic substances identified using the two methods (FD and OSME). The number of substances with an FD ≥6 identified in TN by the FD method was the highest, followed by QM, JM, HY, and GF. The highest number of aromatic components with intensity values >2 was also found in TN, followed by HY and JM. QM and GF had the same number of aromatic substances. Therefore, the identification results obtained using FD and OSME were similar. These were highly consistent in identifying key aromatic substances. 2-Cyclohepten-1-one, ethyl cyclopropanecarboxylate, 1,3,8-*p*-menthatriene, and citral were identified for the first time and regarded as a characteristic aroma substance in mangoes, regardless of analytical method used (FD or OSME), although 1,3,8-*p*-menthatriene and citral were previously detected in lychee fruits [16]. In contrast to previous findings of aromatic substances in mangoes, some potent odorants, such as linalool and isoamyl acetate, were not detected, which might be due to differences in varieties, storage conditions, and extraction techniques. According to sensory analysis, the main aromatic profiles (fruit, sweet, flower, and rosin aromas) of pulp from the five mango cultivars were similar to those identified using the FD and intensity methods, indicating that the intensity method combined with the FD method can accurately illustrate the characteristic aromatic components with high or low intensity.

### 3.4. Odor Activity Values (OAVs) for the Pulps of Five Mango Cultivars

Aromatic analysis techniques such as the FD and intensity methods can effectively analyze the major aromatic compounds in mango pulp, but these cannot accurately reflect the contributions of individual components to the overall aroma characteristics [24]. Therefore, OAVs may be a more accurate scale to evaluate the contribution of volatile substances to fully consider the interactions between the food matrix and aromatic substances. The main component of mango pulp is water, and the calculation of the OAVs of each substance was based on the results of accurate quantitative analysis performed in this study, and the aroma threshold of each compound in water was previously reported [25,26].

According to the literature [19], substances with OAVs >1 contribute to the overall aroma of the sample. Table 3 and Figure 6 show the results of OAV analysis of pulp from five mango cultivars. There were 25 characteristic aromatic components with an OAV ≥ 1 in pulps of five mango cultivars, including two alcohols, seven aldehydes, three esters, 11 terpenes, and two ketones. Terpenes (44%) and aldehydes (28%) were the main aromatic components of mango, of which γ-terpinene had the highest OAV (3.04–10.04), followed by β-phellandrene (2.41–3.41), hexanal (1.10–16.97), and 1-nonanal (5.37–56.2), which were also considered as major aroma-active compounds in Australian mango cultivars. In contrast, although alcohols were the predominant component of all substances (Table 1), these showed minimal contribution (8%), due to their relatively high odor threshold. For instance, the OAV of the highest concentration of (e)-3-hexen-1-ol was only within the range of 0.38–1.21. In addition, 2-cyclohepten-1-one, ethyl cyclopropanecarboxylate, 1,3,8-*p*-menthatriene, and citral were first identified to be useful in aroma activity in mango based on OVA, which coincides with the results of GC-O. 

OAV values of characteristic aromatic substances were also different in different varieties of mango. Table 3 shows 11 substances with OAV ≥1 in JM. The OAV of γ-terpinene was the largest (7.70), followed by 1-hexanol (6.20), γ-octanoic (2.66), phellandrene (2.41), and ethyl cyclopropanecar-boxylate (2.17). In QM, nine aromatic substances with OAV ≥1 were identified, of which 1-nonanal (9.73) had the largest OAV, followed by 1-hexanol (6.58), γ-terpinene (3.37), *p*-cymene (1.83), and β-phellandrene (1.55). GF has nine substances with OAV ≥1, among which 1-nonanal (6.07) has the largest OAV, followed by γ-terpinene (3.04), 1-hexanol (2.55), γ-octanoic (2.36), decanal (1.41), and *p*-cymene (1.05). HY has eight substances with OAV ≥1, among which 1-nonanal (56.2) has the largest OAV value, followed by 1-hexanal (16.79), trans-2-heptenal (5.05), 2-cyclohepten-1-one (4.71), and γ-octanoic (3.76). HY has 17 substances with OAV ≥1, among which γ-terpinene (10.44) has the largest OAV, followed by *p*-cymene (7.19), terpinolene (5.80), 1-nonanal (5.73), 1-hexanol (5.13), and β-phellandrene (4.31).

### 3.5. Comparison of GC-O(FD/OSME) and OAV Aroma-Active Compounds 

The joint analysis revealed 29 components (FD ≥ 6, OSME ≥ 2, OAV ≥ 1) as aroma-active compounds in the pulps of five mango cultivars (Figure 7). A total of 28 components were detected by GC-O (FD/OSME), whereas 25 substances were detected only by OVA. Compounds with high OAVs, such as 1-nonanal (5.73–56.20), ethyl butyrate (1.56–5.40), and heptanal (1.65–1.83), were not detected using GC-O (FD/OSME). Among the components discriminated by all the panelists in GC-O (FD/OSME), the contributions of 2-penten-1-ol, β-pinene, 3-methylcyclohex-3-en-1-one, and 6-methyl-5-hepten-2-one to the overall mango aroma were limited, as their OAVs were ≤0.1. The discrepancies between the two assessments mainly resulted from differences in application principles [19]. The calculations of the OAVs were based on the odor threshold in water instead of the food matrix. For the actual food matrix, the release of aroma is promoted or inhibited by interactions of volatiles with food components [27,28,29]. Therefore, OAV identification may not precisely match the actual results generated using GC-O (FD/OSME). However, biological variations, including respiratory rate and receptor state, may lead to errors in aromas based on GC-O (FD/OSME). This explains why the synergetic use of the two methods is strongly recommended for the identification of aroma-active compounds. In this study, the results of OAV coincided with those of GC-O (FD/OSME) to a certain extent, and the 29 key contributors to the five Chinese mango pulps were thus identified. These included 1-penten-3-one, 2-vyclohepten-1-one, 2-penten-1-ol, hexanal, ethyl cyclopropanecarboxylate, *trans*-2-hexenal, 3-hexenal, 1-hexanol, heptanal, α-pinene, β-pinene, e-2-heptenal, 3-methylcyclohex-3-en-1-one, 6-methyl-5-hepten-2-one, β-myrcene, ethyl butyrate, 2-carene, β-phellandrene, 3-carene, *p*-cymene, d-limonene, (E)-beta-ocimene, γ-terpinene, terpinolene, 1-nonanal, 1,3,8-*p*-menthatriene, citral, decanal, and γ-octanoic. These were all recognized as potent and major aroma contributors to mango pulp flavor. Further investigation showed that 1-hexanol, γ-terpinene, β-phellandrene, terpinolene, ethyl cyclopropanecarboxylate, and γ-octanoic were aroma-active compounds in JM; β-phellandrene, *p*-cymene, d-limonene, decanal, 1-hexanol, 1-nonanal were the most important aromatic substances in QM; 1-nonanal, 2-cyclohepten-1-one, 1,3,8-*p*-menthatriene, hexanal, 2-cyclohepten-1-one, and γ-octanoic were the most important aromatic substances in HY; and 1-hexanol, γ-terpinene, γ-octanoic, β-phellandrene, and 1-nonanal were the most important aromatic substances in GF. TN was significantly higher than in the other four aromatic substances. β-myrcene, 2-carene, β-phellandrene, 3-carene, *p*-cymene, d-limonene, γ-terpinene, terpinolene, 1,3,8-*p*-menthatriene, and decanal were the most important aromatic substances in HY.

## 4. Conclusions

Mango has a pleasing sensory quality and rich nutritional components, and thus it is essential to study the composition of flavor components in mango. A total of 47 volatile compounds were preliminarily identified by GC-MS, which were subsequently classified into alcohols, alkenes, aldehydes, esters, ketones, and ethers. The results of GC-O (FD/OSME) analysis showed that there were 23, 20, 20, 24, and 24 kinds of aromatic components in JM, QM, GF, HY, and TN, respectively. Sensory analysis indicated that the main sensory aroma profiles (fruit, sweet, flower, and rosin aromas) of the pulps of five mango cultivars were consistent with those of identified using the FD and OSME methods, indicating that the intensity method combined with the FD method could accurately reflect the characteristic aromatic components with high or low intensities. Moreover, OAV calculations indicated that there were 11 substances with OAVs ≥1 in JM, nine in QM, nine in GF, eight in HY, and 17 in TN. Analysis of OAV and GC-O(FD/OSME) identified 29 predominant aroma-active compounds (FD ≥ 6, OSME ≥ 2, OAV ≥ 1) in the pulps of five mango cultivars, which included citrus, lemon-like γ-terpinene and β-phellandrene, rosin-like terpinolene, floral, green-like 1-hexanol and γ-octanoic, and fruit-like ethyl cyclopropanecarboxylate in JM. The predominant aroma-active compounds of cucumber, fruity, floral, citrus, green-like cucumber-like β-phellandrene, *p*-cymene, d-limonene, decanal, 1-hexanol, and 1-nonanal were observed in QM. The predominant aroma-active compounds of minty, citrus, green, floral, violet, coffee, cucumber-like 1-nonanal, 2-cyclohepten-1-one, 1,3,8-*p*-menthatriene, hexanal, 2-cyclohepten-1-one, and γ-octanoic were detected in HY. The predominant aroma-active compounds of resin, flower, green, citrus, lemon, cucumber-like 1-hexanol, γ-terpinene, γ-octanoic, β-phellandrene, and 1-nonanal were observed in GF. Light balsam, wood, sweet, rosin, citrus, minty, fruity, citrus, orange-like β-myrcene, 2-carene, β-phellandrene, 3-carene, *p*-cymene, d-limonene, γ-terpinene, terpinolene, 1,3,8-*p*-menthatriene, and decanal were the most important aromatic substances in HY. TN was significantly higher than HY in four other aromatic substances. In addition, 2-cyclohepten-1-one, ethyl cyclopropanecarboxylate, 1,3,8-*p*-menthatriene, and citral were identified to be associated for the first time with aroma activity in mango based on OVA and GC-O(FD/OSME). Hence, this research not only revealed the aroma-active compounds in different mangoes, but also improved our understanding and control of critical aroma parameters in different mango cultivars in China.

## Figures and Tables

**Figure 1 foods-09-00075-f001:**
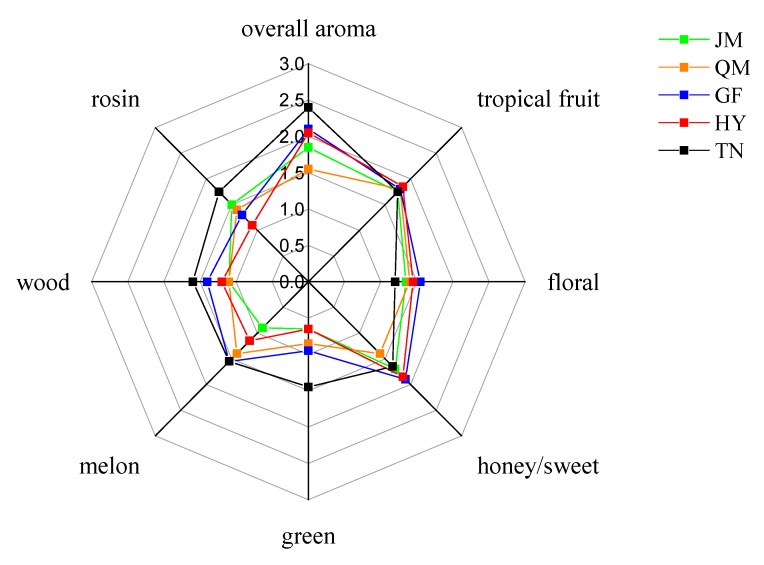
Spider plot for flavor attributes of five varieties mango samples. Jinmang—JM, Qingmang—QM, Hongyu—HY, Guifei—GF, and Tainong—TN.

**Figure 2 foods-09-00075-f002:**
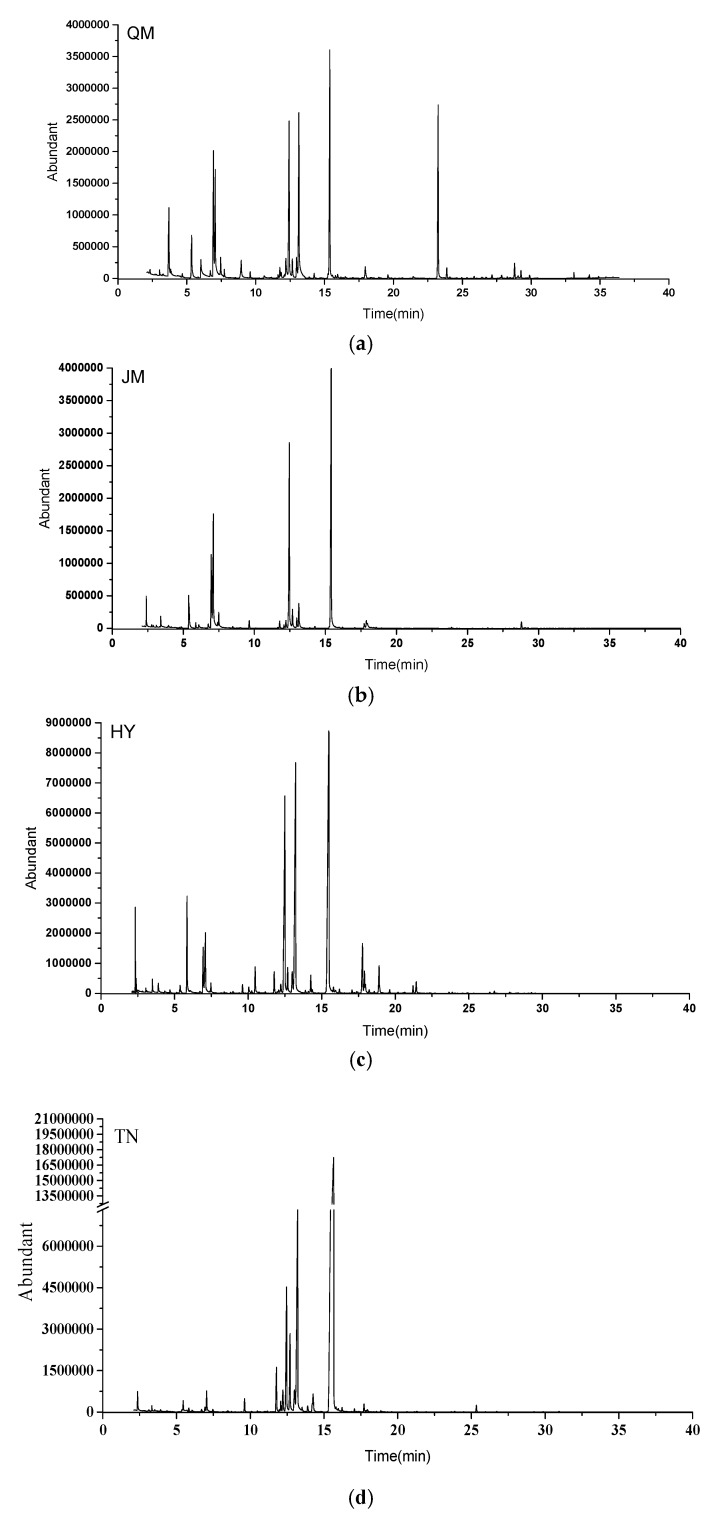
The representative Total ion chromatograms of five varieties mango samples. **a**: QM, **b**: JM, **c**: HY, **d**: TN, **e**: GF.

**Figure 3 foods-09-00075-f003:**
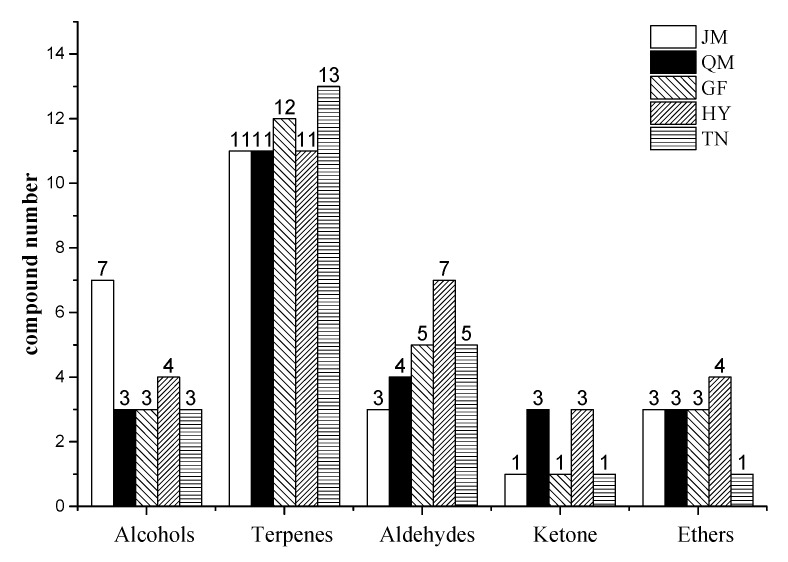
Comparisons of the numbers of volatile compounds detected in different mango samples

**Figure 4 foods-09-00075-f004:**
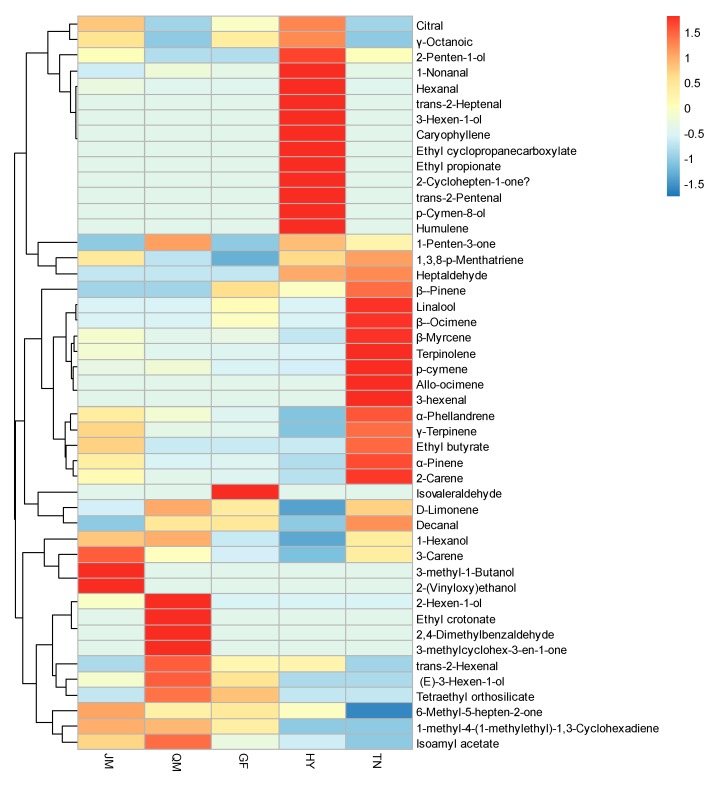
Hierarchical clustering of the all compounds in different mango samples.

**Figure 5 foods-09-00075-f005:**
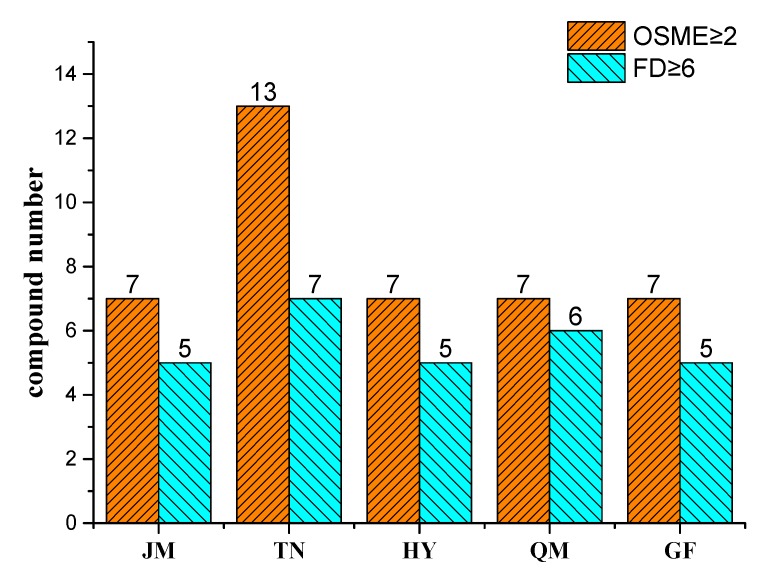
Identification of key aromatic compounds by frequency detection (FD) ≥6 and order-specific magnitude estimation (OSME) ≥2 in five mango pulps.

**Figure 6 foods-09-00075-f006:**
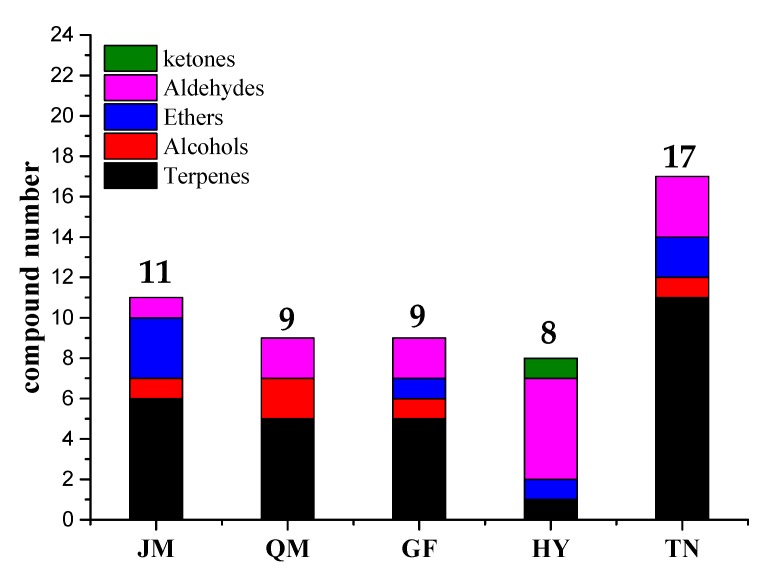
Identification of key aromatic compounds by OVA ≥ 1 in five mango pulps.

**Figure 7 foods-09-00075-f007:**
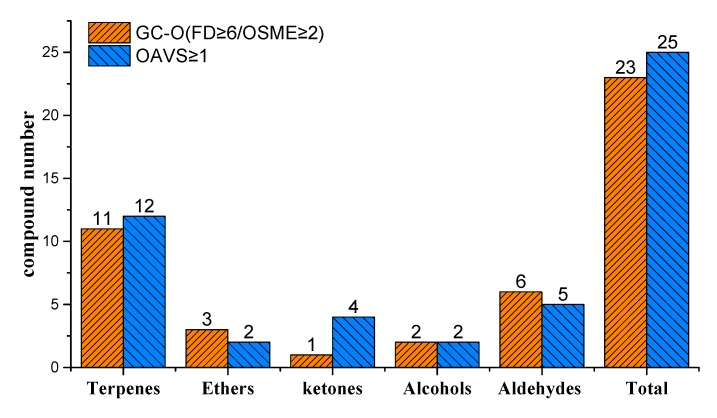
Comparison of GC-O (FD/OSME) and OAVs for aroma-active compounds.

**Table 1 foods-09-00075-t001:** Volatile compounds identified in five different cultivars of Chinese mango samples using SPME-GC/MS.

Code	Compounds	RI ^1^		Identification ^2^	RQCF (fi′) ^3^	Concentration (μg/kg Mango Must) ^4^									
		HP-5	Reference			JM	RSD (%)	QM	RSD (%)	GF	RSD (%)	HY	RSD (%)	TN	RSD (%)
Alcohols															
A1	2-Penten-1-ol	761	769	RI, Std, MS,	1.25	1.92 ^b5^	0.02	- ^8^	-	-	-	5.45 ^a^	0.22	1.87 ^c^	0.07
A2	(E)-3-Hexen-1-ol	844	852	RI, Std, MS,	0.52	41.50 ^c^	0.03	132.56 ^a^	0.12	76.82 ^b^	0.12	-	-	-	-
A3	2-Hexen-1-ol	852	852	RI, Std, MS,	1.09	7.08 ^b^	0.23	35.53 ^a^	0.11	-	-	-	-	-	-
A4	3-Hexen-1-ol	854	857	RI, Std, MS,	4.3	-	-	-	-	-	-	6.59	0.11	-	-
A5	1-Hexanol	856	865	RI, Std, MS,	2.01	55.83 ^b^	0.08	59.24 ^a^	0.13	22.91 ^d^	0.10	7.71 ^e^	0.12	46.14 ^c^	0.17
A6	isoamyl alcohol	921	910	RI, Std, MS,	0.74	38.87	0.03	-	-	-	-	-	-	-	-
A7	p-Cymen-8-ol	945	ND	RI, Std, MS,	11.08	-	-	-	-	-	-	12.23	0.04	-	-
A8	2-(Vinyloxy) ethanol	1012	ND	RI, Std, MS,	3.34	3.49	0.06	-	-	-	-	-	-	-	-
A9	Linalool	1103	1098	RI, Std, MS,	0.80	-	-	-	-	1.92 ^b^	0.07	-	-	6.90 ^a^	0.2
Terpenes															
B1	α-Pinene	918	916	RI, Std, MS,	0.36	4.05 ^b^	0.24	1.76 ^d^	0.17	1.93 ^c^	0.12	1.01 ^e^	0.17	7.38 ^a^	0.04
B2	Allo-ocimene	947	950	RI, Std, MS,	2.56	-	-	-	-	-	-	-	-	0.08	0.12
B3	β--Pinene	958	946	RI, Std, MS,	2.10	-	-	-	-	2.33 ^b^	0.08	1.34_c_	0.17	3.54 ^a^	0.01
B4	β-Myrcene	973	990	RI, Std, MS,	0.43	6.28 ^b^	0.24	3.83 ^d^	0.34	4.5 ^c^	0.07	1.41 ^e^	0.17	22.94 ^a^	0.05
B5	2-Carene	981	995	RI, Std, MS,	0.22	1.60 ^b^	0.31	0.67 ^c^	0.14	0.54 ^d^	0.22	-	-	4.44 ^a^	0.05
B6	Humulene	984	990	RI, Std, MS,	0.81	-	-	-	-	-	-	2.93	0.03	-	-
B7	α-Phellandrene	985	989	RI, Std, MS,	0.50	16.84 ^b^	0.24	10.84 ^c^	0.01	6.90 ^d^	0.07	-	-	30.17 ^a^	0.04
B8	3-Carene	990	1107	RI, Std, MS,	0.25	103.63 ^a^	0.24	60.36 ^c^	0.23	43.32 ^d^	0.07	27.67 ^e^	0.17	70.92 ^b^	0.01
B9	D-Limonene	1007	1026	RI, Std, MS,	0.10	12.4 ^d^	0.24	32.54 ^a^	0.23	26.2 ^c^	0.07	2.56 ^e^	0.17	28.87 ^b^	0.04
B10	*p*-cymene	1005	1016	RI, Std, MS,	1.11	27.43 ^c^	0.08	33.85 ^b^	0.14	19.44 ^d^	0.06	15.57 ^e^	0.09	132.96 ^a^	0.04
B11	β-Caryophyllene	1018	ND	RI, Std, MS,	0.64	-	-	-	-	-	-	2.18	0.17	-	-
B12	1,3-Cyclohexadiene, 1-methyl-4-(1-methylethyl)	1021	1030	RI, Std, MS,	0.18	4.96 ^a^	0.07	4.84 ^b^	0.02	3.42 ^c^	0.06	-	-	-	-
B13	γ-Terpinene	1036	1045	RI, Std, MS,	0.68	15.44 ^b^	0.16	6.73 ^c^	0.19	6.08 ^d^	0.26	1.10 ^e^	0.07	20.88 ^a^	0.25
B14	β-Ocimene	1050	1060	RI, Std, MS,	1.52	-	-	-	-	6.97 ^b^	0.08	-	-	9.11 ^a^	0.04
B15	Terpinolene	1066	1065	RI, Std, MS,	0.19	147.07 ^b^	0.24	59.42 ^c^	0.23	43.5 ^d^	0.07	24.90 ^e^	0.17	811.60 ^a^	0.04
B16	1,3,8-*p*-Menthatriene	1086	1110	RI, Std, MS,	1.37	11.92 ^c^	0.18	3.79 ^d^	0.19	-	-	13.58 ^b^	0.16	16.38 ^a^	0.14
Aldehydes															
C1	Hexanal	792	800	RI, Std, MS,	1.30	4.94 ^b^	0.07	-	-	-	-	75.54 ^a^	1.50	-	-
C2	Isovaleraldehyde	837	ND	RI, Std, MS,	2.47	-	-	-	-	1.77	0.17	-	-	-	-
C3	trans-2-Hexenal	840	844	RI, Std, MS,	2.20	8.38 ^c^	0.02	180.60 ^a^	0.26	84.03^b^	0.15	85.64 ^b^	1.50	4.11 ^d^	0.26
C4	3-hexenal	857	847	RI, Std, MS,	11.55	-	-	-	-	-	-	-	-	44.13	0.12
C5	Heptanal	888	899	RI, Std, MS,	0.07	-	-	-	-	-	-	1.65 ^b^	1.50	1.83 ^a^	0.44
C6	trans-2-Heptenal	940	957	RI, Std, MS,	1.35	-	-	-	-	-	-	65.65	1.50	-	-
C7	1-Nonanal	1080	1140	RI, Std, MS,	0.45	-	-	1.46 ^b^	0.16	0.91 ^c^	0.28	8.43 ^a^	1.50	0.86 ^c^	0.26
C8	(E,Z)-2,6-Nonadienal	1152	1152	RI, Std, MS,	0.64	-	-	-	-	-	-	0.66	0.10	-	-
C9	E-2-Nonenal	1162	1160	RI, Std, MS,	1.87	-	-	7.60	0.09	-	-	-	-	-	-
C10	Decanal	1175	1205	RI, Std, MS,	6.05	-	-	8.48 ^c^	0.06	8.45 ^c^	0.15	-	-	12.46 ^a^	0.04
C11	Citral	1236	1242	RI, Std, MS,	0.02	0.18 ^b^	0.28	-	-	0.09 ^c^	0.05	1.23 ^a^	0.36	-	-
ketones															
D1	1-Penten-3-one	652	687	RI, Std, MS,	0.05	-	-	0.52 ^a^	0.56	-	-	0.47 ^b^	0.28	0.31 ^c^	0.22
D2	2-Cyclohepten-1-one	673	ND	RI, Std, MS,	1.50	-	-	-	-	-	-	0.66	0.14	-	-
D3	3-methylcyclohex-3-en-1-one	967	986	RI, Std, MS,	3.15	-	-	0.62	0.17	-	-	-	-	-	-
D4	6-Methyl-5-hepten-2-one	969	985	RI, Std, MS,	0.42	0.39 ^a^	0.05	0.28 ^c^	0.22	0.30 ^b^	0.25	0.23 ^d^	0.32	-	-
Ethers															
E1	Ethyl propionate	765		RI, Std, MS,	1.10	-	-	-	-	-	-	6.39	0.11	-	-
E2	Ethyl cyclopropanecarboxylate	808	755	RI, Std, MS,	0.26	-	-	-	-	-	-	0.12	0.55	-	-
E3	Ethyl crotonate	876	802	RI, Std, MS,	1.60	-	-	9.83	0.05	-	-	-	-	-	-
E4	Isoamyl acetate	879	876	RI, Std, MS,	1.37	5.31 ^b^	0.09	7.45 ^a^	0.22	2.21 ^c^	0.19	1.28 ^d^	0.14	-	-
E5	Tetraethyl orthosilicate	923	880	RI, Std, MS,	0.06	-	-	0.12^a^	0.54	0.09 ^b^	0.93	-	-	-	-
E6	Ethyl butyrate	978	ND	RI, Std, MS,	0.84	1.17 ^b^	0.06	-	-	-	-	-	-	1.80 ^a^	0.23
E7	γ-Octanoic	1222	994	RI, Std, MS,	2.48	18.60 ^b^	0.21	-	-	16.49 ^c^	0.20	26.32 ^a^	0.11	-	-

^1^ Retention index of volatile compounds on HP-5 columns according to equation proposed [30]; reference: comparing linear retention indices (LRI) on columns (HP-5) in the literature. “ND” not detected in literature. ^2^ Method of identification: RI, retention index (HP-5) in agreement with literature value; Std, confirmed by authentic standards; MS, mass spectrum comparisons using NIST14 library; ^3^ RQCF (fi′) equals the ratio of quantitative factor of identified components standards to that of internal standard (ethyl hexanoate). ^4^ Concentrations are expressed in nanograms per milliliter of mango must, with ethyl hexanoate as the internal standard, and data listed are the means of three assays ± RSDs (%); all RSDs were <15%. ^5^ Values in total data with different letters are significantly different (*p* < 0.05). ^8^ “-” not detected in samples.

**Table 2 foods-09-00075-t002:** GC-O identified aroma-active compounds in mango samples with the method of aromatic intensity and frequency.

No.	Compounds	RI (Calculate) ^A^		RI (Reference) ^B^		Aroma description ^C^	Identification ^D^	Aromatic Intensity ^E^										Frequency ^F^				
		HP-5	Wax	HP-5	Wax			JM	RSD (%)	TN	RSD (%)	HY	RSD (%)	QM	RSD (%)	GF	RSD (%)	JM	TN	HY	QM	GF
1	1-Penten-3-one	652	ND	687 [6]	ND	mushroom	R, S, M, O	0	0	2.25	10.00	1.67	5.99	1.33	11.28	0	0	0	5	4	8	0
2	2-Cyclohepten-1-one	673	ND	ND	ND	coffee-like	R, S, M, O	0	0	0	0	2.00	0.50	0	0	0	0	0	0	5	0	0
3	2-Penten-1-ol	761	1316	769 [31]	1305 [6]	grassy, tea	R, S, M, O	2.00	0.01	1.25	0.80	1.00	1.00	0	0	0	0	4	2	7	0	0
4	Hexanal	792	1061	800 [32]	1088 [32]	grass, tallow, fat	R, S, M, O	1.83	0.49	2.2	0.91	2.00	1.00	0	0	1.80	0.56	5	2	2	0	2
5	Ethyl cyclopropanecarboxylate	808	ND	ND	ND	fruity	R, S, M, O	1.50	0.23	1.23	0.20	1.30	0.90	0	2.11	0.56	0	6	6	8	0	8
6	trans-2-Hexenal	840	1202	844 [32]	1192 [32]	green, leaf	R, S, M, O	1.50	0.23	1.67	6.95	1.67	3.78	2.00	0.50	2.00	0.50	3	5	4	5	4
7	3-hexenal	857	ND	857 [6]	ND	grass	R, S, M, O	1.27	0.67	0	0	1.36	3.78	1.60	8.13	1.43	0.45	3	1	7	8	7
8	3-Hexen-1-ol, (E)-	844	1356	852 [31]	1343 [31]	grassy, tea	R, S, M, O	1.00	0.01	0	0	0	0	1.33	0.75	1.75	5.71	4	2	0	2	3
9	1-Hexanol	856	1360	865 [32]	1362 [32]	resin, flower, green	R, S, M, O	1.60	0.28	1.83	2.73	1.23	0.56	2.12	0.34	2.13	0.67	5	2	4	4	3
10	Heptanal	888	1174	899 [6]	1184 [32]	fruity	R, S, M, O	0	0	1.5	0.67	1.67	2.34	1.00	1.00	0	0	0	5	4	0	0
11	α-Pinene	918	1032	916 [31]	1056 [23]	pine-like	R, S, M, O	1.10	0.34	1.80	10.00	2.20	20.00	1.00	1.00	1.5	0.67	2	4	2	2	2
12	trans-2-Heptenal	940	1300	957 [31]	1307 [31]	grassy, irritant	R, S, M, O	0	0	0	0	0	0	1.25	1.60	1.56	0.9	0	2	1	3	3
13	Allo-ocimene	947	1125	ND	1135 [1]	floral	R, S, M, O	0	0	0	0	1.00	1.00	0	0	0	0	0	0	3	0	0
14	β-Pinene	958	1113	946 [23]	1108 [23]	grass	R, S, M, O	0	0	1.56	0.80	1.23	1.50	1.67	0.60	0	0	0	5	4	7	0
15	3-methylcyclohex-3-en-1-one	967	ND	986 [1]	ND	grass	R, S, M, O	0	0	0	0	0	0	2.67	2.25	0	0	0	1	0	3	0
16	6-Methyl-5-hepten-2-one	969	ND	985 [23]	ND	fruit, grass	R, S, M, O	1.67	0.32	2.00	0.50	1.33	4.51	1.56	0.78	1.75	3.43	2	4	2	2	3
17	β-Myrcene	973	1137	990 [32]	1158 [32]	light balsam, wood	R, S, M, O	1.80	1.20	2.00	0.50	0	0	0	0	0	0	4	6	0	0	0
18	Ethyl butyrate	978	1070	994 [31]	1076 [31]	fruity, apple	R, S, M, O	1.23	5.79	1.67	6.95	0	0	0	0	0	0	3	5	0	0	0
19	2-Carene	981	ND	995 [23]	ND	sweet, rosin	R, S, M, O	1.89	0.98	2.20	2.27	0	0	1.54	0.56	1.23	0.44	5	6	0	4	2
20	β-Phellandrene	985	1166	989 [31]	1171 [31]	citrus like	R, S, M, O	4.10	1.34	3.53	6.67	3.00	0.33	2.10	0.90	2.50	0.67	7	6	2	5	4
21	3-Carene	990	1148	1007 [31]	1153 [31]	sweet, rosin	R, S, M, O	1.50	0.87	2.00	0.50	1.14	0.34	1.45	0.76	1.23	1.56	3	4	2	2	2
22	*p*-Cymene	1005	1275	1026 [32]	1274 [32]	citrus, green	R, S, M, O	1.00	0.29	8.70	0.45	1.67	6.95	2.00	15.0	1.80	6.67	5	6	6	6	6
23	D-Limonene	1028	1205	1030 [32]	1208 [32]	citrus-like, sweet	R, S, M, O	0	0	2	0.5	0	0	2.00	0.50	0	0	0	5	0	4	0
24	(E)-beta-ocimene	1050	1241	1048 [33]	1250 [33]	floral and green	R, S, M, O	2.33	0.01	0	0	2.50	0.40	0	0	0	0	3	0	3	0	0
25	γ-Terpinene	1057	1249	1060 [33]	1245 [33]	citrus, lemon	R, S, M, O	2.33	0.02	2.50	1.60	2.33	1.72	1.67	6.95	2.00	2.50	8	8	5	6	8
26	Terpinolene	1066	1275	1065 [31]	1276 [31]	rosin-like	R, S, M, O	2.30	0.02	2.28	2.19	2.20	3.18	1.33	0.75	2.33	9.31	6	8	7	8	7
27	1-Nonanal	1080	1328	1104 [31]	1349 [31]	cucumber-like	R, S, M, O	1.67	0.26	1.50	3.33	0	0	0	0	1.40	7.14	5	4	0	0	3
28	1,3,8-*p*-Menthatriene	1086	ND	1110 [34]	ND	minty-like	R, S, M, O	1.00	0.01	2.00	0.50	2.20	0.45	0	0	2.25	4.44	5	4	4	0	4
29	(E,Z)-2,6-Nonadienal	1152	1460	1152 [31]	1469 [6]	fresh, green, cucumber	R, S, M, O	0	0	0	0	1.67	0.60	0	0	0	0	0	0	2	0	0
30	E-2-Nonenal	1162	1416	1160 [31]	1436 [31]	cucumber-like	R, S, M, O	1.20	0.01	0	0	0	0	0	0	0	0	2	0	0	0	0
31	Citral	1236	1700	1242 [34]	1679 [33]	lemon-like	R, S, M, O	2.00	0.01	0	0	0	0	0	0	0	0	2	0	0	0	0
32	Decanal	1175	1835	1205 [34]	1846 [34]	fruity, citrus, orange	R, S, M, O	0	0	3	0.33	0	0	3.00	0.33	0	0	0	2	0	2	0
33	γ-Octanoic	1222	1721	1236 [31]	1733 [31]	floral, violet	R, S, M, O	2.00	0.01	0	0	1.50	2.00	0	0	2.00	0.50	7	0	4	0	3

^A^ Retention index of volatile compounds on columns (HP-5 and WAX) according to equation proposed [30]; “ND” not detected in samples. ^B^ RI (reference): comparing linear retention indices (LRI) on columns (HP-5 and WAX) in the literature. ^C^ Odor note perceived at the sniffing port. ^D^ Method of identification: RI, retention index (HP-5) in agreement with literature value; Std, confirmed by authentic standards; MS, mass spectrum comparisons using NIST14 library. ^E^ Aromatic intensity, the data listed are the means of three assays ± RSDs (%); all RSDs were <15%. ^F^ Aroma frequency.

**Table 3 foods-09-00075-t003:** Concentrations and calculations of odor activity values (OAVs) of the important aroma-active compounds in mango samples.

No	Compounds	Threshold ^a^ (μg kg^−1^)	Source ^b^	OAV ^c^				
				JM	QM	GF	HY	TN
1	1-Penten-3-one	1.00 [35]	mango	0	0.52	0	0.47	0.31
2	2-Cyclohepten-1-one	0.14 [35]	new	0	0	0	4.71	0
3	2-Penten-1-ol	720.00 [23]	mango	<0.01	0	0	<0.01	<0.01
4	Hexanal	4.50 [23]	mango	1.10	0	0.29	16.79	0.40
5	Ethyl cyclopropanecarboxylate	0.12 [23]	new	2.17	0	0	0	1.00
6	trans-2-Hexenal	400.00 [23]	mango	0.02	0.45	0.21	0.21	0.01
7	3-hexenal	550.00 [23]	mango	0	0	0	0	0.08
8	3-Hexen-1-ol, (E)-	110.00 [23]	mango	0.38	1.21	0.70	0	0
9	1-Hexanol	9.00 [26]	mango	6.20	6.58	2.55	0.86	5.13
10	Heptanal	1.00 [23]	mango	0	0	0	1.65	1.83
11	α-Pinene	6.00 [35]	mango	0.68	0.29	0.32	0.17	1.23
12	trans-2-Heptenal	13.00 [35]	mango	0	0	0	5.05	0
13	Allo-ocimene	140.00 [35]	mango	0	0	0.07	0.04	0.10
14	β-Pinene	100.00 [23]	mango	0	0	0	<0.01	0
15	3-methylcyclohex-3-en-1-one	7.00 [36]	mango	0	0.09	0	0	0
16	6-Methyl-5-hepten-2-one	50.00 [23]	mango	<0.01	<0.01	<0.01	<0.01	0
17	β-Myrcene	20.00 [23]	mango	0.31	0.19	0.23	0.07	1.15
18	Ethyl butyrate	0.75 [23]	mango	1.56	0	0	0	2.40
19	2-Carene	4.00 [23]	mango	0.40	0.17	0.14	0	1.11
20	β-Phellandrene	7.00 [23]	mango	2.41	1.55	1.00	0	4.31
21	3-Carene	50.00 [23]	mango	2.07	1.21	0.87	0.55	1.42
22	*p*-Cymene	18.50 [23]	mango	1.48	1.83	1.05	0.84	7.19
23	D-Limonene	26.00 [23]	mango	0.48	1.25	1.01	0.10	1.11
24	(E)-beta-ocimene	6.70 [23]	mango	0	0	1.04	0	1.36
25	γ-Terpinene	2.00 [23]	mango	7.72	3.37	3.04	0.55	10.44
26	Terpinolene	140.00 [23]	mango	1.05	0.42	0.31	0.18	5.80
27	1-Nonanal	0.15 [23]	mango	0	9.73	6.07	56.20	5.73
28	1,3,8-*p*-Menthatriene	6.80 [23]	Lychee	1.75	0.56	0	2.00	2.41
29	(E,Z)-2,6-Nonadienal	4.50 [23]	mango	0.57	0	0	0	0.02
30	E-2-Nonenal	50.00 [36]	mango	0	0.15	0	0	0
31	Citral	1.00 [23]	Lychee	0.18	0	0.09	1.23	0
32	Decanal	6.00 [23]	mango	0	1.41	1.41	0	2.08
33	γ-Octanoic	7.00 [35]	mango	2.66	0	2.36	3.76	0

^a^ OT odor threshold in water (ppb) found in the newly determined and taken from the literature. ^b^ Source: It indicates substances found in the related literature for mangoes and litchis; New: first identified to be useful in aroma activity in mango. ^c^ An OAV was calculated by dividing the concentration of an odorant by its orthonasal odor threshold.

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
