# Peer review of "Aromatic Characterization of Mangoes (Mangifera indica L.) Using Solid Phase Extraction Coupled with Gas Chromatography–Mass Spectrometry and Olfactometry and Sensory Analyses"

_foods, 2020, doi:10.3390/foods9010075_

Round 1

Reviewer 1 Report

The article by Liu et al. is an original work about key aroma compounds in mango (Mangifera indica L.). I have just a couple of minor points that require the attention of the authors:

Comparison with current data in the literature. There is a conspicuous number of papers on aroma compounds in literature. The innovative content of the presented work is not well addressed. The authors generically say that the “Scientific information on aroma constituents and sensory characteristics of different cultivars mangoes is rather scarce” but this should be better stated and quantified. I suggest to comment and add a table reporting which compounds were already characterized and the references for these previous characterizations, and for these compounds when and which thresholds are known (or add this info in existing tables). Discuss the different methodologies and which are the advantages/improvements and the need for this work.

The text should be carefully revised since there are several typos: g., line 44: a small numbers, line 45 sensory characteristics of different cultivars mangoes is rather scarce, line 271: the highest number of aroma components with intensity values greater than 2 were also…, etc. (very often the space after the comma is missing)

Author Response

Dear reviewer,

Many thanks to the reviewers for these precious comments concerning my manuscript entitled “Aroma characterization of mangoes(Mangifera indica L.)using solid phase extraction coupled with gas chromatography–mass spectrometry and olfactometry and sensory analysis”. These comments are very valuable and helpful for revising and improving my paper. Appended to this letter is our point-by-point response to the comments raised by the reviewers with red writing. Please see the attachment.

Sincerely,

Haocheng LIU

Reviewer 2 Report

Concerning the manuscript entitled "Aroma characterization of mangoes(Mangifera indica L) using solid phase extraction coupled with gas chromatography–mass spectrometry and olfactometry and sensory analysis”, the topic contains some interesting information and direct logical arguments seem to be satisfactorily developed. Moreover, a lot of valuable results are presented.

The paper is very pleasant to read although language should be at some places improved: English should be edited by a native English speaker.

Author Response

Dear reviewer,

Many thanks to the reviewers for these precious comments concerning my manuscript entitled “Aroma characterization of mangoes(Mangifera indica L.)using solid phase extraction coupled with gas chromatography–mass spectrometry and olfactometry and sensory analysis”. These comments are very valuable and helpful for revising and improving my paper. Appended to this letter is our point-by-point response to the comments raised by the reviewers with red writing.Please see the attachment.

Sincerely,

Haocheng LIU

Reviewer 3 Report

Dear Authors,

I have thoroughly reviewed the subject matter of the manuscript.
Although the work appears to have been carried out with care, in my judgment, the significance of the manuscript seems not to be broad enough and does not provide a sufficient advance in our understanding of Aroma characterization of fresh fruits.

In particular, authors do not provide any information about the chemical composition of mangoes utilized for the characterization. Furthermore, neither information was provided about the real maturity index of the fruits or about all the other factors that could influence the final aroma of fruits (i.e. agronomical practices, terroir, storage conditions adopted in postharvest period, etc.).

Also, the statistical analysis of sensorial data seems lacking.

In this context the real manuscript topics seems to be limited at the characterization of the aroma of five different varieties of mangoes by the merging of different analytical procedures but it is not possible to conclude if the differences highlighted were related to the varieties or to other factors not well defined.

Finally, while the manuscript is well written and understandable, in my opinion it appears completely lacking in novelty and also the its scientific soundness is very poor.

Regards

Author Response

(The authors gave the same response as above.)

Round 2

Reviewer 3 Report

Dear Authors,

I have no other comments.

Regards